# Classification of Polish Natural Bee Honeys Based on Their Chemical Composition

**DOI:** 10.3390/molecules27154844

**Published:** 2022-07-28

**Authors:** Barbara Pacholczyk-Sienicka, Grzegorz Ciepielowski, Jakub Modranka, Tomasz Bartosik, Łukasz Albrecht

**Affiliations:** Institute of Organic Chemistry, Faculty of Chemistry, Lodz University of Technology, Zeromskiego 116, 90-924 Lodz, Poland; grzegorz.ciepielowski@p.lodz.pl (G.C.); jakub.modranka@p.lodz.pl (J.M.); tomasz.bartosik@p.lodz.pl (T.B.); lukasz.albrecht@p.lodz.pl (Ł.A.)

**Keywords:** chemometrics, honey analysis, qNMR

## Abstract

The targeted quantitative NMR (qNMR) approach is a powerful analytical tool, which can be applied to classify and/or determine the authenticity of honey samples. In our study, this technique was used to determine the chemical profiles of different types of Polish honey samples, featured by variable contents of main sugars, free amino acids, and 5-(hydroxymethyl)furfural. One-way analysis of variance (ANOVA) was performed on concentrations of selected compounds to determine significant differences in their levels between all types of honey. For pattern recognition, principal component analysis (PCA) was conducted and good separations between all honey samples were obtained. The results of present studies allow the differentiation of honey samples based on the content of sucrose, glucose, and fructose, as well as amino acids such as tyrosine, phenylalanine, proline, and alanine. Our results indicated that the combination of qNMR with chemometric analysis may serve as a supplementary tool in specifying honeys.

## 1. Introduction

The global honey market was valued at over 8 billion U.S. dollars in 2021 and is expected to grow annually by 5.2% (Compound Annual Growth Rate, CAGR 2022–2030) [1]. The European Union is the world’s second biggest producer of honey after China [2]. According to data of The Research Institute of Horticulture Apiculture division in Pulawy, there are around 82100 beekeepers and approximately 1.77 million bee families in Poland; however, the production of honey dropped from 22 thousand tons in 2018 to about 7–10 thousand tons in 2021 [3]. The production does not cover the demand; thus, in 2021, about 32,184 tons of honey were imported (mainly from Ukraine and China), while only 981 tons were exported to Spanish, German, and French markets [2]. For comparison, in 2019, Poland exported almost 17 thousand tons of honey; thus, a significant decrease in honey production was observed. The limited supply of honey as well as the increased demand for this product caused by the spread of the SARS-CoV-2 coronavirus resulted in an increase in honey prices. Therefore, domestic honey is perceived this year as a deficit commodity and enjoys a greater interest among consumers, especially in the case of monofloral honeys due to their multi-faceted beneficial properties. The production of monofloral honeys is also of interest to beekeepers owing to the opportunity to compete with low-priced polyfloral honeys as well as honeys imported from abroad [3].

Lately, a rising interest in the nutritional properties and precious quality of this product has been observed. It is caused by the change in our lifestyles and growing nutritional awareness of consumers, who want to maintain a healthy lifestyle. In comparison with other sweeteners, honey is the most expensive product; therefore, consumers want to have 100% certainty that they are buying a safe and full-value product. An increasing amount of customer attention has been focused on the quality of honey and its adulteration, as well as its compliance with its label description relaying to important scientific tasks. The scale of the problem is demonstrated by the fact that the number of papers related to honey authenticity and classification is constantly growing (Figure 1).

In this context, honey authenticity monitoring, safety and quality checks, as well as label compliance have become crucial scientific goals [4]. The composition and quality parameters for honey were clearly defined by European legislations [5,6]. Among these parameters, the contents of sugar, water, nonwater-soluble fraction, conductivity, free acidity, diastase activity, and hydroxymethylfurfural were determined. European regulations did not determine the amino acids content [7]. The composition and properties of bee honey are connected with its geographical and floral origin, as well as bee-keeping season: spring or summer, environmental factors, and the treatment of beekeepers such as the temperature and storage time [8,9]. The melissopalynological analysis is one of the main methods for authentication of monofloral honey; however, this method is time-consuming and often unreliable [10]. To determine the unifloral origin of honeys, organoleptic analysis is also applied; however, it is a very subjective method. Due to the fact that honey consists of about 200 compounds such as sugars, water, proteins, organic acids, minerals, phenolic compounds, pigments, and volatile compounds [9,11], a lot of different analytical techniques have been applied to assess the quality and origin of honey. Among methods used for the determination of honey authenticity and composition are high-performance liquid and gas chromatography [7,12,13,14], atomic adsorption spectrometry (AAS), inductively coupled plasma-mass spectrometry (ICP-MS) [15,16], matrix-assisted-laser-desorption/ionization time-of-flight mass spectrometry MALDI TOF MS [17], Raman spectroscopy [18], infrared spectroscopy (NIR, FT-IR) [19,20], IRMS [21], and NMR spectroscopy [21,22,23,24,25,26,27].

Due to the large amounts of analytical data obtained from spectroscopic and chromatographic methods, the application of more advanced calculations such as the chemometric approach is required. Among the most popular chemometric methods in honey analysis are principal component analysis (PCA), partial least squares analysis (PLS), or linear discriminant analysis (LDA) [15,16,17,18,19,20,21,22,23,24,25,26,27]. These methods reduce the multidimensionality of data and allow the discovery of the possible relations and/or existing differences between different groups of samples such as authentic and adulterated or different types of honey.

The aim of this study was to classify the botanical origin of four types of Polish honeys: lime (*Tilia* L.), rape (*Brassica napus* L.), acacia (*Robina pseudoacacia*), and multiflorous by means of the quantitative NMR (qNMR) approach combined with chemometric techniques. To avoid processed and commercially available honey, for this purpose, fresh samples directly from beekeepers were purchased. All honey samples were collected during two beekeeping seasons. Moreover, six samples were also bought in local supermarkets, for separate analysis and comparison with our database set. The parameters of the method such as linearity, and intra-day and inter-day repeatability were thoroughly examined.

## 2. Results and Discussion

A total of 34 honey samples of different botanical origins were analyzed. Among these, 28 Polish samples were chosen within four equally represented groups (7 samples each). Three groups were selected as monofloral products against the last group including seven multifloral honeys. Monofloral honeys are distinguished according to the unique botanical species available for the bees working out the honey products, specifically clustered in lime (*Tilia* L.), rape (*Brassica napus* L.), and acacia (*Robina pseudoacacia*) samples. Assignment of compound resonances was based on analysis of one-dimensional ^1^H, two-dimensional correlation spectroscopy (COSY), and heteronuclear single-quantum coherence spectroscopy (HSQC) NMR spectra. The collected data were compared with those from the literature and with the spectra of the metabolites database recorded on a Bruker Avance II Plus 700 MHz spectrometer. NMR spectra for honey specimens were very similar and signals of four groups of compounds were identified. The first group was comprised of carbohydrates (mainly glucose—Glc, fructose—Fru, sucrose—Suc, and arabinose—Ara), and the second included amino acids (tyrosine—Tyr, alanine—Ala, phenylalanine—Phe, and proline—Pro). Moreover, carboxylic acids and their derivatives, mainly lactic acid—Lac, formic acid—FA, dimethyl succinic acid, 4-chlorobenzoic acid, and 5-(hydroxymethyl)furfural—HMF, were found in the spectra. Ethanol was also identified in some of the rape honey samples. Representative ^1^H NMR spectra of the lime, rape, acacia, and multifloral honey samples are shown in Appendix A. The main compounds characterizing investigated samples with their diagnostic ^1^H signals, chemical shifts δ_H_, and multiplicities are displayed in Appendix A.

Prior to the classification of the botanical origin of Polish honey samples, the quantification of main constituents was performed. Unfortunately, for most of the identified compounds, direct quantification by means of integration is not possible, due to the overlapping signals. A large number of overlapped signals for all isomeric forms of sugars was observed with the highest peak intensities in the range of the sugar chemical shift belonging to glucose and fructose. As a consequence, eight compounds for which at least one good resolved signal in all analyzed spectra was identified were chosen. Selected peaks included: Ala (3H, d, 1.47 ppm), Pro (1H, m, 2.3–2.39 ppm), Glc (1H, d, 4.64 ppm), Fru (1H, d, 4.11 ppm), Suc (1H, d, 5.44 ppm), Tyr (1H, d, 6.91 or 7.18 ppm), Phe (2H, d, 7.28 ppm or 1H, t, 7.37 ppm), and hydroxymethylfurfural (1H, d, 6.67 ppm or 1H, s, 9.45 ppm). For all mentioned compounds, the calibration curves were performed to check the linear response between the integrals and the concentration of each compound. Good correlation coefficients (R > 0.99) were observed for each calibration graph. Except for the calibration curve for Glc, which is described in Section 3.3.1, the remaining curves are provided in Appendix A.

### 2.1. Sugar Profile of Polish Honeys

The sugar profile of fructose, glucose, and sucrose depends on the botanical and geographical origin of honey; thus, it is affected by factors such as seasonal climate, processing, and storage [9,28,29]. The average sugar content in different varieties of natural honey and the ratio of fructose to glucose are given in Table 1.

Among Polish honey of different botanical origin, the highest content of glucose was observed for rape honey at 38.74 ± 3.25 g/100 g, while the lowest was observed in acacia honey at 27.62 ± 1.73 g/100 g. In the case of fructose, the highest average content was determined in multiflorous honeys, while lime honeys were characterized by the lowest concentration of this sugar. The lowest sucrose concentration was observed for rape, while the highest was observed in multiflorous honeys. Borawska et al. also studied the sugar content in different varieties of Polish honeys, and in the case of rape honey (*n* = 5), the average glucose concentration of 39.5 ± 2 g/100 g was observed, while the fructose content was 38.2 ± 1 g/100 g [30]. Hao et al. also reported that acacia honeys were characterized by Glc and Fru contents at levels of 27.2 ± 2.6 and 39.4 ± 1.3 g/100 g, respectively [31]. Those values are in accordance with our own results. The sugar profile for German rape honey was determined by Ohmenhaeuser et al. [22], and slightly lower concentrations of glucose and fructose (34.9 and 37.9 g/100 g, respectively) were reported. Escuredo et al. demonstrated that among all studied honey samples, the lowest concentration of fructose was observed for rape and dandelion honeys. In the case of these honeys, the glucose content is higher than fructose, which results in rapid crystallization [28]. In our studies, the concentrations of both sugars were comparable, while in Borawska et al., the glucose content was slightly higher than the fructose content [30]. Our results indicated the highest average fructose content for foreign nectars (49.82 ± 17.74 g/100 g) and for Polish multiflorous honeys (43.51 ± 1.22 g/100 g), which may result from the botanical diversity of pollen present in honey. These data are consistent with results described by Borawska et al. [30]. Moreover, our results regarding sugar are in accordance with another study showing similar concentration levels of the same sugars, except sucrose in acacia honeys, which was found in larger concentrations by Zalewski et al. [32].

Among all studied samples, foreign commercial multiflorous honeys were characterized by the highest average sucrose content of 3.76 ± 1.38 in the range from 2.96 to 6.51 g/100 g. In the case of one sample, the sucrose content was above the maximum limit of 5 g/100 g as required by Codex Alimentarius 2001 [5]. Higher levels of sucrose from 0.2 to 2.3 g/100 g were also reported by Escudero et al. in all varieties of the Spanish honeys [28].

Rodriguez et al. demonstrated that the fructose-to-glucose ratio (F/G) may serve as an indicator of honey flavor due to the fact that fructose is sweeter than glucose and sucrose. Thus, honeys with the highest fructose-to-glucose ratio were sweetest [33]. In our studies, the highest F/G ratio was observed for acacia (1.52 ± 0.10), while the lowest was for rape honeys (1.01 ± 0.09). A similar tendency was described by Zalewski et al., where the F/G ratio for acacia equaled 1.55, for lime 1.12, for floral 1.08, and for rape 0.94 [32]. In the work of Borawska et al., the lowest F/G ratio was also observed for rape (0.97 ± 0.03), while for lime and multiflorous honeys, they were found at 1.09 ± 0.1 and 1.16 ± 0.1, respectively. The highest F/G ratio may be related to the time of production of honey by bees (the later the honey is produced, the higher the F/G ratio) [30].

### 2.2. Amino Acids Profile and HMF Content in Polish Honeys

Apart from sugars, examples of amino acids composition as indicators of the botanical origin of floral honeys were reported in the literature [34,35,36]. The presence of protein and amino acids in the chemical composition of honey is connected with animal and vegetal sources, such as the pollen. According to the literature review, a total of 26 amino acids were found in honey samples, whereas their amount depends on their nectar or honeydew origin [37]. Among the amino acids present in honey are proline, alanine, glutamic acid, valine, threonine, leucine, phenylalanine, methionine, cysteine, isoleucine, aspartic acid, tyrosine, tryptophan, asparagine, glycine, histidine, and serine [34,37,38]. However, the most common amino acids in honey samples are proline, alanine, tyrosine, phenylalanine, isoleucine, leucine, and glutamic acid [9]. The predominant amino acid of honey is proline, which is an indicator of maturity of honey, because its content decreases during storage [35]. Moreover, proline concentration may serve as an additional parameter for botanical origin characterization [39]. In the case of honeys adulterated by sugar, the proline and HMF content are lowered [37]. Proline content is used to estimate the quality of honey and may indicate adulteration with sugar according to the Council Directive 2001/110/EC, where the 18 mg/100 g minimum value of proline is accepted as the limit concentration for pure honey [6].

In our studies, a significantly lower proline concentration was measured in acacia honeys of 11.34 ± 2.94 mg/100 g in comparison to other varieties of honeys (Table 2). This result may indicate the immaturity of investigated honeys or may indicate that the bees were fed with sucrose. However, there are reports of a lower value of proline content in some types of honeys. Kowalski et al. [14] and Janiszewska et al. [7] also reported lower proline levels in acacia samples (21.77 ± 6.59 and 22.57 ± 4.39 mg/100 g, respectively) in comparison with lime and rape honeys. A significantly lower concentration of proline was also found in acacia honeys from Hungary at 25.2 ± 3.8 mg/100 g [37] and Italy at 22.9 ± 3.4 mg/100 g [39]. Among of Polish honeys, the highest concentration of Pro was found in lime honey at 64.77 ± 9.99 mg/100 g, while results obtained by Borawska et al. indicated the average proline content at 40 ± 15 mg/100 g for these kinds of nectars [30]. Similar Pro contents in lime honeys (37.67 ± 13.43 mg/100 g) have been reported by Kowalski et al. [14]. In turn, the average proline content for Italian and Hungarian lime honeys equals 38.8 ± 2.5 and 69.7 ± 24.8 mg/100 g, respectively [37,39]. The proline content for rape samples was estimated at 26.83 ± 3.06 mg/100 g, which is consistent with mean values provided in the literature for Polish rape honeys at 28.3 ± 2 mg/100 g [30]. However, lower values of proline for rape honeys were reported by Kowalski et al. (16.3 ± 1.07 mg/100 g) [14]. In rape honeys from Hungary, the Pro content was determined at 37.7 ± 6 mg/100 g [37]. Polish multiflorous honey samples were characterized by a Pro concentration equal to 41.04 ± 4.75 mg/100 g, and this value is also consistent with data obtained for this kind of honey by Borawska et al. (41.7 ± 8 mg/100 g) [30]. In the case of studies conducted by Kowalski et al. for multiflorous honeys, the proline content was determined at 29.26 ± 13.98 mg/100 g [14]. Proline quantities of floral honeys from Hungary, Croatia, France, and Spain were about 54.2 ± 14.3, 60.1 ± 1.0, 42.9 ± 2.0, and 67.41 ± 16.17 mg/100 g, respectively [37,40].

In the case of foreign multiflorous honey samples, the Pro content was estimated at 67.55 ± 4.71, which is consistent with the mean values of 68.9 ± 1.6 mg/100 g for floral EU and not EU honeys reported in the literature [37].

Apart from proline, the acacia honeys showed a higher concentration of tyrosine at 7.21 ± 0.66 mg/100 g (Table 2), which is in accordance with results reported by Janiszewska et al. [7]. In our studies, the concentration of Phe in acacia samples was 1.81 ± 0.14 mg/100 g. Similar results were obtained by Kowalski et al. (1.71 ± 1.08 mg/100 g) [14]. Lime honeys showed, in addition to Pro, a high amount of Phe (4.46 ± 0.25 mg/100 g) and comparable amounts of Ala and Tyr (Table 2). In turn, rape honeys were characterized with a high content of phenylalanine (3.78 ± 0.23 mg/100 g). Other significant amino acids in this group of honey were tyrosine and alanine in descending order from 2.11 to 1.44 mg/100 g. In multiflorous honey, we observed (apart from proline) a high content of Phe (8.09 ± 0.99 mg/100 g), Tyr (3.51 ± 0.39 mg/100 g), and Ala (2.30 ± 0.13 mg/100 g). In an amino acids analysis conducted by Kowalski et al., tyrosine was not detected in all studied samples, while average Ala concentrations did not exceed 1.8 mg/100 g [14]. In general, Phe is the second most abundant amino acid immediately after Pro in honey samples, which was confirmed by our results.

Foreign multiflorous honey samples indicated similar concentrations of Phe and Ala to Polish multiflorous honeys and a higher amount of Tyr.

Moreover, the analysis for hydroxymethylfurfural (HMF), which is the parameter used for the evaluation of honey freshness and/or overheating, was performed. HMF is a compound that changes its concentration as a function of time. According to the European regulation, an HMF content of 40 mg/kg is accepted as the maximum value for honey (or 80 mg/kg in the case of honey from tropical regions) [41,42]. The concentration of HMF is determined in honey samples during quality control, but it does not provide any information about botanical and/or geographical origin [43].

The highest concentration of HMF in our studies was measured for rape honeys (16.18 ± 1.39 mg/kg), while the lowest was for acacia honeys (0.83 ± 0.08 mg/kg, Table 2). This tendency was also observed in studies performed by Zalewski et al., but the mean concentrations are at the level of 0.2 mg/kg for rape and 0.07 mg/kg for acacia [32]. The amount of 10 mg/kg of HMF was found in rape honeys from Germany [22]. In turn, Murtabegovic et al. studied eight samples of acacia honey of different geographical origin, in which HMF concentrations at the level ranging from 0.02 to 10 mg/kg were observed [44]. All analyzed Polish honey samples met the requirements for the HMF content defined in the Regulation, providing that the samples were fresh and unprocessed.

### 2.3. One-Way ANOVA

The significant differences in concentrations of sugars, amino acids, and HMF between different honey samples were determined using one-way analysis of variance (ANOVA). The eight quantified compounds showed statistically significant differences that could be used to specify honey botanical origin. These results are presented in Figure 2, Figure 3 and Figure 4. The sucrose values showed significant differences in the means for all types of honey (Figure 2c). The mean content value for Glc allowed the discrimination of almost all types of honey, except multiflorous and lime, where the differences were not statistically significant (Figure 2a). As shown in Figure 2b, the concentration of fructose seemed to be the clearest indicators for discriminating honey samples of different botanical origin, in addition to acacia from multiflorous honey. 

The one-way ANOVA test of Tukey’s comparison was also performed for amino acids and HMF concentrations (Figure 3 and Figure 4). On the basis of the proline and tyrosine average contents, the differentiation between all types of honey proved possible. Apart from the rape and lime honeys, the concentrations of Phe were statistically significant for all analyzed nectars. Alanine showed a statistically significant difference only in acacia and rape honey. Additionally, analysis of variance for HMF indicated statistically different concentrations for all types of honey except acacia and lime samples.

### 2.4. Multivariate Data Analysis

Principal component analysis (PCA) concerns the explanation of the variance structure of a set of variables through linear combinations of the variables (principal components, PCs). The PCs are orthogonal variables, which are obtained by multiplying the original correlated variables with the eigenvector (coefficients). The participation of the original variables in the PCs is shown in the loadings plot (axes within +1 and −1), where high absolute values indicate variables with important meaning for a considered PC and a low absolute value corresponds to a variable of minor importance. The loading plot indicated the compounds that were associated with the similarity/dissimilarity between the observations of the study [16].

PCA was applied to the autoscaled data matrix in order to observe the trends and similarities between analyzed honey samples. The chemical concentration of selected compounds: sugars, amino acids, and HMF, were chosen as input data for analysis. As shown in Figure 5a, two PCs accounted for 65.46% of the total variation in accordance with PC1 = 46.36% and PC2 = 19.10%. The loading plot of a principal component indicated original variables, which contributed significantly to those principal components (Figure 5b). PC1 with an eigenvalue of 3.3093 was positively correlated with Glc, Fru, HMF, Ala, and Tyr, while it negatively correlated with Suc, Pro, and Phe:PC1 = 0.3817 × Glc + 0.4899 × Fru − 0.3999 × Suc − 0.3958 × Pro − 0.1571 × Phe + 0.4268 × HMF + 0.3931 × Tyr + 0.0555 × Ala(1)

In the case of PC2 with the eigenvalue of 1.3399, a positive correlation was observed for Fru, Suc, Phe, Ala, and Tyr, while it was negative for Pro, Glc, and HMF:PC2 = 0.0933 × Fru + 0.4287 × Suc − 0.3211 × Pro + 0.4817 × Phe − 0.1053 × HMF − 0.1971 × Glc + 0.4758 × Tyr + 0.4996 × Ala(2)

Despite the fact that PCA belonged to an unsupervised method, sufficient results for most of the analyzed types of honey were obtained. As shown on the scores plot in Figure 5a, lime honeys (red square) were located at negative scores for PC1 and PC2; thus, lime honeys were distinguished from the group of acacia and multiflorous honeys, which were located at positive scores of PC2, as well as from rape honeys located at positive scores of PC1. The highest negative loadings of proline were mainly responsible for this differentiation. Multiflorous and acacia honeys were not clearly separated from each other, which means that these honeys have similar characteristics. The highest positive loadings of sucrose and Phe on PC2 were associated with multiflorous and acacia honeys. Both kinds of honey were characterized by similar concentrations of Suc and extremely different concentrations of Phe. Rape honeys presented a good separation from all other samples, which is correlated with the positive loadings of Glc and HMF on PC1 and Ala, and Tyr and Fru on PC2. In comparison to the means of the rape honeys with other types of honeys, we found that the means of HMF and Glc were highest, while those for Ala, Suc, and Tyr were lowest for rape honey. A clear separation was observed for foreign multiflorous honey, which was characterized by the highest concentration of Suc, Fru, and HMF.

Our results showed that the sugar profile in combination with the amino acid composition of honeys allowed clear distinction of the botanical origin of Polish honeys. Kowalski et al. [14] and Janiszewska et al. [7] indicated that on the basis of amino acids profile alone, the differentiation of botanical origin of Polish honeys was impossible due to high variability. However, according to several authors, the composition of amino acids may serve as a suitable method to determine the botanical origin of floral honey [34,35,45]. An analysis conducted for Finnish honeys of different botanical origin demonstrated that the most significant loadings to discriminate the honeys corresponded to Glc and Fru, while low-magnitude loadings corresponded to amino acids [46]. As claimed by Ohmenhaeuser et al. [22] and Lolli et al. [47], glucose and fructose play the key role in the classification of honey samples. 

## 3. Materials and Methods

### 3.1. Honey Samples

A total of 34 honey samples of different botanical origins were analyzed. Among them, 28 Polish honey samples were purchased directly from certified beekeepers within one month from honey harvesting, including 7 lime honeys (July), 7 rape honeys (first half of May), 7 acacia honeys (May/June), and 7 multifloral honeys. These samples were collected between 2020 and 2021 from domestic apiaries located in central and northern Poland. Moreover, six samples were also bought in local supermarkets, for separate analysis and comparison with our database set. The origin of each honey sample was declared either by beekeepers or in the case of commercial honeys by the seller. Most of commercial samples were declared as multifloral blends of EU honey and non-EU honey. The samples are described in Table 3.

### 3.2. Sample Preparation

All samples were prepared in the same way according to the protocol described by Ralli et al. [48]. A portion of honey samples (150 mg) were weighted and dissolved with 0.6 mL of D_2_O (100% deuterated). Sodium-3′-trimethylsilyl-2,2,3,3-d_4_ propionate (TSP) was used as an internal reference for chemical shift (0 ppm) and for quantitative analysis. The TSP concentration was matched as 0.25 mg/mL. The samples (600 µL) were transferred to 5 mm NMR tubes and were allowed to stand at rest to fully equilibrate before NMR experiments. The pH of the samples was in the range of 4.0–4.3.

### 3.3. NMR Experiments

All spectra were acquired using a Bruker Avance II Plus 16.4 T spectrometer (Bruker BioSpin, Rheinstetten, Germany) operating at an ^1^H frequency (700.16 MHz). The instrument was equipped with a 5 mm Z-gradient broadband decoupling inverse probe. All experiments were performed at 300 K. The standard proton spectra with water presaturation (*zgcppr* pulse program, Bruker, Karlsruhe, Germany) were acquired with a calibrated 90° pulse for 256 scans collecting 64 K data points over a spectral width of 12 ppm. The repetition time of 6 s was matched to ensure complete magnetization recovery. Quantitative ^1^H NMR spectra were recorded within 26 min per sample. An exponential line broadening of 0.05 Hz was applied to raw data prior to Fourier transformation. The TSP peak at 0 ppm was used as a chemical shift standard. 

#### 3.3.1. Calibration Curve 

Quantitative accuracy requires the correct repetition time (T_r_), which is defined as the time from the application of an excitation pulse to the next pulse, and it includes the relaxation delay and the acquisition time [49]. To ensure complete magnetization recovery, the T_R_ value should be calculated as a multiple (five to seven times) of the longest T_1_ in the sample. The quality of an analytical method is dependent on the linearity of the calibration curve. To check the linearity of the proposed method, ^1^H NMR spectra for 8 series of glucose solutions (0.48, 0.96, 1.92, 4.21, 8.43, 17.50, 33.71, and 66.82 g/100 g) were recorded with two different repetition times: T_r_ = 6 s and 8.5 s. Each sample was analyzed in triplicate in random order. Linear relationships with very good correlation coefficients (R = 0.9999 and R = 0.99999) were observed when the absolute concentration of glucose was plotted against that calculated from integrals. The results are shown in Figure 6 and in Appendix A. Satisfactory results were obtained for both calibration curves; thus, for further measurements, the repetition time of 6 s was selected to minimize the time of experiments. Moreover, as a criterion of acceptance for the analytical method, we assumed a recovery between 98 and 102%; therefore, we rejected the repetition time of 8.5 s, where a recovery above 102% was observed (see Appendix A).

#### 3.3.2. Inter-Day and Intra-Day Repeatability

The intra-day and inter-day repeatability for the ^1^H NMR experiments were calculated by performing measurements on the same day and on the different days, respectively. Spectra were recorded for the same sample of glucose solution (1.92 g/100 g). As shown in Table 4, satisfactory results of the statistical analysis were obtained.

Moreover, a similar analysis was performed for the rape honey sample, where the concentration of Glc was estimated at 31.5147 g/100 g (Table 5).

### 3.4. Quantification of Sugars and Amino Acids

Qualitative and quantitative analyses were performed for the spectral region from −0.1 to 8.0 ppm. Spectra were phased and referenced to TSP manually with the Topspin software (version 3.2, Bruker, Germany). All spectral regions were individually corrected over the integrated regions using a fifth-order baseline function. Sugars and amino acids were identified either by the analysis of one-dimensional ^1^H and two-dimensional spectra or by spiking experiment.

### 3.5. Statistical Analysis

Statistical analysis was performed using OriginPro 2020 9.7.0.188 (OriginLab Corporation). One-way analysis of variance (ANOVA) was performed on sugars, amino acids, and 5-(hydroxymethyl)furfural contents to determine significant differences in compound levels between Polish honey samples. The Tukey test was performed to compare differences among the mean values at a confidence level of *p* < 0.05. The variation in the data was explored by principal component analysis (PCA), which was used for unsupervised pattern recognition, allowing the observation of trends and similarities between samples. Sugars, amino acids, and HMF contents were chosen as input data for PCA analysis. Prior to PCA analysis, data were autoscaled by subtracting their mean values and dividing by the standard deviation. Each honey sample was characterized by 8 variables, i.e., glucose, fructose, sucrose, proline, phenylalanine, tyrosine, alanine, and HMF, which resulted in a 34 × 8 data matrix with the honey types as rows and the variables as columns.

## 4. Conclusions

The classification of 34 honey samples of different botanical origin by means of eight descriptors was performed. The NMR quantitative approach was used to evaluate the concentration of sugars and amino acids in rape, lime, acacia, and multiflorous honey samples. High correlation coefficients were obtained for each calibration curve, which confirmed the very good linear response within the concentration range of analyzed compounds. For pattern recognition, principal component analysis (PCA) was performed and good separations between all honey samples were obtained. Our results indicated that the combination of qNMR with chemometric analysis may serve as a supplementary tool in specifying honeys. The presented method is fast: the total experiment time was 32 min and was characterized by a simple sample preparation. However, to construct a good classification model for routine analysis, the number of floral honey types, as well as the total number of investigated samples, should be increased; therefore, further studies on the application of NMR spectroscopy in such studies are ongoing in our laboratory. 

## Figures and Tables

**Figure 1 molecules-27-04844-f001:**
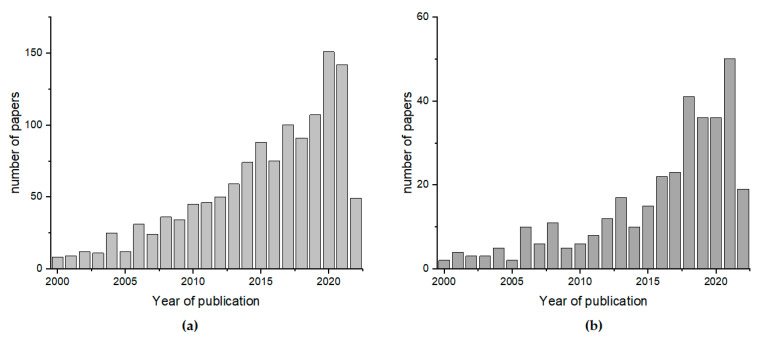
The number of papers published between 2000 and 2022 with the key words (**a**) “honey classification” and (**b**) “honey authentication” according to Scopus (www.scopus.com, accessed on 13 June 2022).

**Figure 2 molecules-27-04844-f002:**
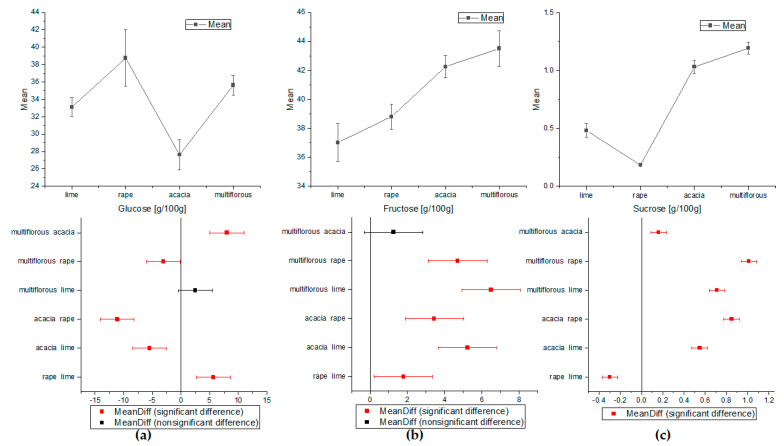
The comparison of the average means of sugars content obtained by ANOVA: (**a**) for glucose, (**b**) for fructose, and (**c**) for sucrose (Tukey test, *p* < 0.05).

**Figure 3 molecules-27-04844-f003:**
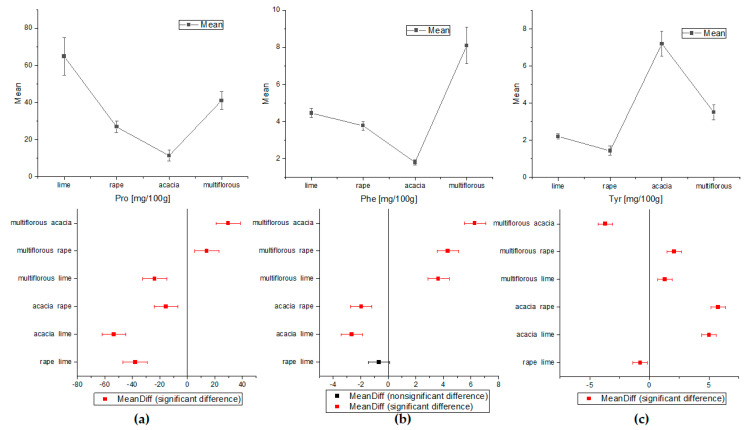
The comparison of the average content of amino acids obtained by ANOVA (**a**) for proline, (**b**) for phenylalanine, and (**c**) for tyrosine (Tukey test, *p* < 0.05).

**Figure 4 molecules-27-04844-f004:**
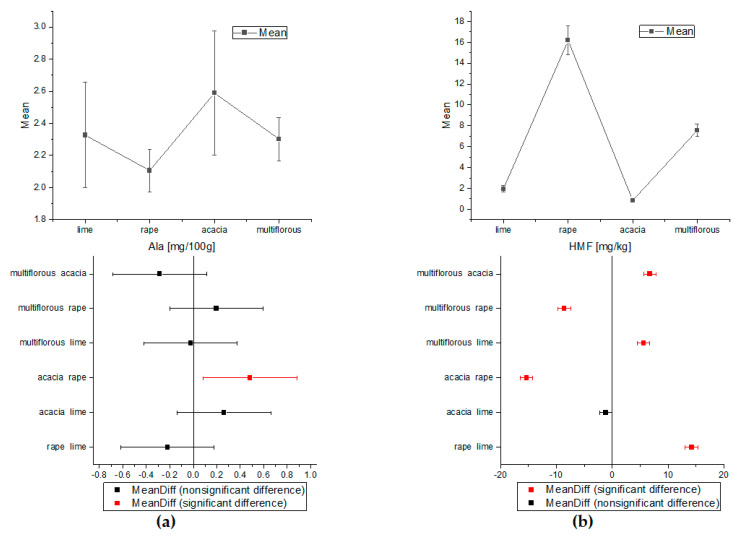
The comparison of the average content of Ala and HMF obtained by ANOVA (**a**) for alanine and (**b**) for HMF (Tukey test, *p* < 0.05).

**Figure 5 molecules-27-04844-f005:**
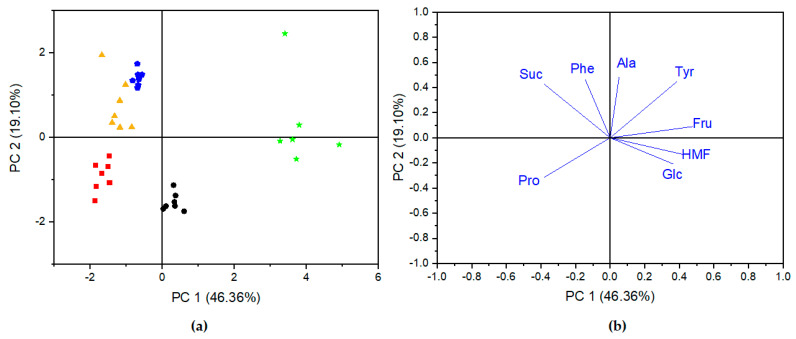
PCA scores plot (**a**) and the corresponding loading plot (**b**) of different honey samples based on the concentration of quantified compounds. Black circle—rape honeys, red square—lime honeys, yellow triangle—multiflorous honeys, blue pentagon—acacia honeys, and green star—foreign multiflorous honeys.

**Figure 6 molecules-27-04844-f006:**
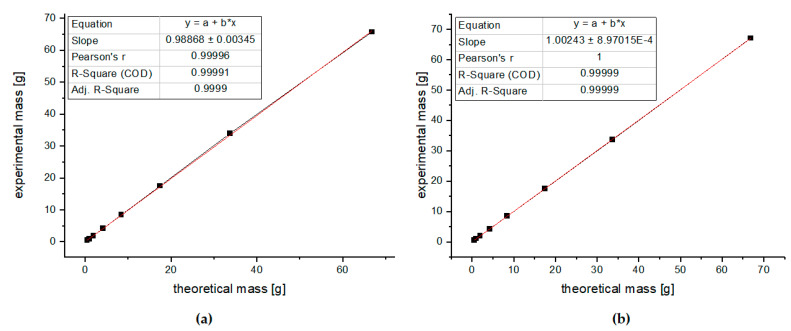
Linearity of the method (**a**) with repetition time of 6 s and (**b**) with repetition time of 8.5 s.

**Table 1 molecules-27-04844-t001:** Comparison of average sugar content in different varieties of honey.

Variety of Honey	Mean ± SD [g/100g]			
	Glc (Min–Max)	Fru (Min–Max)	Suc (Min–Max)	Fru/Glc Ratio
Lime	33.11 ± 1.09 (31.41–33.71)	37.02 ± 1.30 (34.86–38.44)	0.48 ± 0.06 (0.42–0.61)	1.11 ± 0.06 (1.01–1.22)
Rape	38.74 ± 3.25 (37.51–45.28)	38.81 ± 0.87 (37.92–40.03)	0.18 ± 0.004 (0.18–0.19)	1.01 ± 0.09 (0.84–1.14)
Acacia	27.62 ± 1.73 (25.56–30.34)	42.26 ± 0.76 (41.38–43.09)	1.03 ± 0.06 (0.98–1.13)	1.52 ± 0.10 (1.39–1.62)
Multiflorous	35.63 ± 1.13 (34.07–37.74)	43.51 ± 1.22 (41.99–45.24)	1.19 ± 0.05 (1.12–1.27)	1.23 ± 0.07 (1.11–1.32)
Foreign commercial multiflorous	37.32 ± 10.98 (31.41–59.67)	49.82 ± 17.74 (38.44–85.47)	3.76 ± 1.38 (2.96–6.51)	1.32 ± 0.09 (1.18–1.43)

**Table 2 molecules-27-04844-t002:** Comparison of average amino acids and 5-HMF content in different varieties of honey.

Variety of Honey	Mean ± SD [mg/kg]	Mean ± SD [mg/100 g]			
	5-HMF (min-max)	Ala (min-max)	Pro (min-max)	Tyr (min-max)	Phe (min-max)
Lime	1.96 ± 0.32 (1.59–2.37)	2.33 ± 0.33 (1.94–2.83)	64.77 ± 9.99 (52.73–77.17)	2.21 ± 0.15 (2.00–2.38)	4.46 ± 0.25 (4.24–4.82)
Rape	16.18 ± 1.39 (14.71–18.53)	2.11 ± 0.13 (1.91–2.28)	26.83 ± 3.06 (22.03–31.02)	1.44 ± 0.25 (1.16–1-81)	3.78 ± 0.23 (3.30–3.97)
Acacia	0.83 ± 0.08 (0.73–0.95)	2.59 ± 0.39 (1.81–2.97)	11.34 ± 2.94 (8.27–16.72)	7.21 ± 0.66 (6.48–8.06)	1.81 ± 0.14 (1.66–2.04)
Multiflorous	7.57 ± 0.59 (6.91–8.42)	2.30 ± 0.13 (2.11–2.44)	41.04 ± 4.75 (31.21–44.31)	3.51 ± 0.39 (3.01–4.03)	8.09 ± 0.99 (6.81–9.28)
Foreign multiflorous	12.17 ± 0.85 (11.13–13.32)	2.35 ± 0.10 (2.27–2.54)	67.55 ± 4.71 (62.66–74.17)	5.03 ± 0.62 (4.27–5.98)	7.56 ± 0.71 (6.74–8.44)

**Table 3 molecules-27-04844-t003:** Origin and type of honey samples with their denotations.

Honey Samples	Number of Samples	Origin	Symbol ^1^
Lime	7	Beekeeper	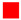
Rape	7	Beekeeper	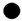
Acacia	7	Beekeeper	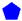
Multiflorous	7	Beekeeper	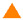
Foreign multiflorous	6	Commercial	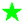

^1^ PCA analysis.

**Table 4 molecules-27-04844-t004:** The intra-day and inter-day repeatability (average value, SD; coefficient of variation, CV) of the applied method for the determination of Glc concentration in standard sample.

Experiment Number	Intra-Day Repeatability	Inter-Day Repeatability
1	1.9198	1.9176
2	1.9224	1.9188
3	1.9211	1.9184
4	1.9207	1.9214
5	1.9178	1.9192
6	1.9216	1.9204
7	1.9179	1.9231
8	1.9194	1.9174
**Average**	**1.9201**	**1.9195**
**SD**	**0.0017**	**0.0020**
**CV%**	**0.0872**	**0.1026**

**Table 5 molecules-27-04844-t005:** The intra-day and inter-day repeatability (average value, SD; coefficient of variation, CV) of the applied method for the determination of Glc concentration in rape honey sample.

Experiment Number	Intra-Day Repeatability	Inter-Day Repeatability
1	37.5147	37.5289
2	37.5157	37.5282
3	37.5161	37.5167
4	37.5149	37.5187
5	37.5152	37.5193
6	37.5168	37.5235
7	37.5156	37.5288
8	37.5164	37.5282
**Average**	**37.5157**	**37.5240**
**SD**	**0.0007**	**0.0052**
**CV%**	**0.0019**	**0.014**

## Data Availability

The data presented in this study are available in the article and in the Appendix A.

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
