# Peer review of "Classification of Polish Natural Bee Honeys Based on Their Chemical Composition"

_molecules, 2022, doi:10.3390/molecules27154844_

Round 1

Reviewer 1 Report

Dear Editor

Respected Authors

The submitted manuscript “1817056” concerns a study based on the NMR quantification (qNMR) of eight main metabolites in honeys; the subsequent statistical analysis is claimed to be able to distinguish three mono-floral varieties against one multifloral class.  The paper is probably containing several scientific information supported by the experimental analytical methods which should not be wasted; however we think that several key issues should be resolved before considering it as possible publication. This reviewer lists below his concerns about the manuscript to be considered together with recommendations possibly coming from other reviewers’.

-       The entire text needs some restyling and maybe a mother tongue reviewer. My humble opinion is to change – Line 10 abstract “..used to determine the chemical profile of different types of Polish honey samples, featured by varable content of main sugars, free aminoacids and hydroxymethyl furfural.”– Line 39 “Lately, it has been observed a rising interests in the nutritional properties and precious quality of this products”. – line 45 “”description relays to important scientific tasks.” – line 51 “In this context, honey authenticity monitoring, safety and quality check as well as label compliance become crucial scientific goals. [4]” (if I got the point) –Line 95, every carbohydrate owns a three letter code so please add it also to arabinose and why not also to the other orphan metabolites-Line 321“….honeys, allowed to clearly distinguish botanical origin of Polish samples”

-       Unfortunately, the general format of §2 Results and §3 Methods (by the way, please place §3 instead of §4 all over paragraphs and sub paragraphs) forces us to tell out first some “experimental information” therefore it is necessary to begin §2 with something like “A total of 34 honey samples of different botanical origin were analyzed. Among these, 28 Polish samples were chosen within four equally represented groups (7 samples each). Three groups were selected as unifloral (maybe monofloral?) products against the last group including seven multifloral honeys. Unifloral honeys are distinguished according to the unique botanical species available for the bees working out the honey products, specifically clustered in lime (Citrus aurantifolia), rape (Brassica nopus) and acacia (Robina pseudoacacia) samples.” Please check the botanical names which should be clear because of the ambiguity of common names.

-       According to lines  154-163 and lines 226-232, we notice that the rough bisector of second and fourth quadrant in figure 5 is strongly related with sucrose-glucose-fructose and HMF and also we remind that these all can be due to the storage time and conditions. This is getting the suspicion that the effective separation among the samples is biased by some different “fate” undergone by different samples. Please add something to defend the unbiased clustering in PCA. Maybe also around line 333 authors should specify the season of purchase from beekeepers and shelf management of beekeepers.

-       §3.2 According to the sample preparation authors just use deuterated solvent without pH regulation. Did they detect resonance shifts? Did they ever measure pH and differences in pH? How

-       §3.3 NMR experiments: it should be the main part of the paper giving crucial information but it is missing several important points. Line 353 zgcppr is not just a water presaturated spectrum, indeed it is also endowed with a pre-acquisition spin-echo sequence probably useful to filter off high molecular weight molecules. Please specify this in the text along with the spinlock time and cycles (tau and big tau in bruker should be d9 and d9*nl)- Line 355: provided that measurement at different repetition time add information to the system, it is at least once useful to run an inversion recovery experiment to experimentally measure T1 for most of the metabolites.- In conclusion it will be also necessary to summarize the total experimental time (as for any analytical method); I suppose it should be around 30 minutes and adding the sample insert and shimming, we could estimate something like 35 minutes but authors should write it down.

-       §3.3.2 Repeatability and reproducibility. According to the exact definition, reproducibility is whether the experiment can be reproduced entirely (with different instrument). According to what it is written the instrument is not changed and maybe the operator neither. If it is the case, probably we should write something like “inter-day and intra-day repeatability” by knowing nothing about the reproducibility. Please answer even though you disagree.    

-       Line 168 “26 amino acids….” Of course including also non alpha proteic aminoacids. Maybe in the supplementary part authors should at least mention those 26 compounds and possibly the NMR signals/assignment beyond table S1

-       Table S1 formic acid is compound #15 but some other compounds should maybe added

-       For discussion in §2.1 and §2.2 many references are mentioned; however by a simple biblio search we notice https://doi.org/10.1021/ac0484979 and https://doi.org/10.1021/jf072763c as related researches. Probably authors should be aware of these studies, but it is not mandatory to mention those

Reviewer 2 Report

The authors developed an analytical method to determine floral origin of honey. By using qNMR and chemometrics, four types of honey could be discriminated successfully. The experiments were designed correctly and the manuscript was well written. However, similar methods have been applied for honey analysis and authors were not successful to distinguish the difference and advantages. Therefore, the manuscript is not suitable for publication in this form. 

Reviewer 3 Report

It is an interesting manuscript, about the use of NMR analysis for the determination of a component in honeys. This study use a simple procedure in a NMR spectrometer, avoiding the tedious and time-consuming chemical methods and other techniques as GC, GC-MS, HPLC & HPLC-MS. It is a great application of NMR technique in honey analysis, a very important task nowadays. The results of this well-done NMR studies proved that this method was a specific, quickly procedure for the analysis of 34 different honeys.  

It is a good manuscript, it should be necessary to improve the text, just only the size of the figures 2, 3 & 4 to make easy to read the text of the figures before publish.

Thanks for the good work!

Round 2

Reviewer 1 Report

Dear Editor Respected Authors

I see that the other reviewers have opposite opinion about this paper and i disagree on my own. Usually I hate to call a second revision because it seems a persecution to authors, so I just recommend a second typing/text revision whose responsibility is entirely on authors. It is not necessary a further reply to this message 

Reviewer 2 Report

The authors successfully revised the manuscript as the reivewers suggested.